# Circular RNAs Modulate Cancer Hallmark and Molecular Pathways to Support Cancer Progression and Metastasis

**DOI:** 10.3390/cancers14040862

**Published:** 2022-02-09

**Authors:** Aliaksandr A. Yarmishyn, Afeez Adekunle Ishola, Chieh-Yu Chen, Nalini Devi Verusingam, Vimalan Rengganaten, Habeebat Aderonke Mustapha, Hao-Kai Chuang, Yuan-Chi Teng, Van Long Phung, Po-Kuei Hsu, Wen-Chang Lin, Hsin-I Ma, Shih-Hwa Chiou, Mong-Lien Wang

**Affiliations:** 1Department of Medical Research, Taipei Veterans General Hospital, Taipei 112, Taiwan; yarmishyn@gmail.com (A.A.Y.); aaishola01@ym.edu.tw (A.A.I.); jerry791204@gm.ym.edu.tw (C.-Y.C.); nalinidv.verusingam.md07@nycu.edu.tw (N.D.V.); vimalan91@1utar.my (V.R.); habeebat094.md08@nycu.edu.tw (H.A.M.); kevin1985336@gm.ym.edu.tw (H.-K.C.); andrea.chi@nycu.edu.tw (Y.-C.T.); phungvanlong@hmu.edu.vn (V.L.P.); shchiou@vghtpe.gov.tw (S.-H.C.); 2Taiwan International Graduate Program in Molecular Medicine, National Yang Ming Chiao Tung University and Academia Sinica, Taipei 112, Taiwan; 3Institute of Pharmacology, School of Medicine, National Yang Ming Chiao Tung University, Taipei 112, Taiwan; 4Centre for Stem Cell Research, Faculty of Medicine and Health Sciences, Universiti Tunku Abdul Rahman, Kajang 43000, Malaysia; 5Postgraduate Programme, Department of Preclinical Sciences, Faculty of Medicine and Health Sciences, Universiti Tunku Abdul Rahman, Kajang 43000, Malaysia; 6School of Medicine, National Yang Ming Chiao Tung University, Taipei 112, Taiwan; pkhsu@vghtpe.gov.tw; 7Division of Thoracic Surgery, Department of Surgery, Taipei Veterans General Hospital, Taipei 112, Taiwan; 8Institute of Biomedical Sciences, Academia Sinica, Taipei 115, Taiwan; wenlin@ibms.sinica.edu.tw; 9Department of Neurological Surgery, Tri-Service General Hospital, National Defense Medical Center, Taipei 114, Taiwan; tsghns01@ndmctsgh.edu.tw; 10Genomic Research Center, Academia Sinica, Taipei 112, Taiwan; 11Institute of Food Safety and Health Risk Assessment, School of Pharmaceutical Sciences, National Yang-Ming Chiao Tung University, Taipei 112, Taiwan

**Keywords:** circular RNA, cancer progression, metastasis, biomarker

## Abstract

**Simple Summary:**

Circular RNAs (circRNA) are a type of RNA molecule of circular shape that are now being extensively studied due to the important roles they play in different biological processes. In addition, they were also shown to be implicated in disease such as cancer. Cancer is a complex process which is often defined by a combination of specific processes called cancer hallmarks. In this review, we summarize the literature on circRNAs in cancer and classify them as being implicated in specific cancer hallmarks.

**Abstract:**

Circular RNAs (circRNAs) are noncoding products of backsplicing of pre-mRNAs which have been established to possess potent biological functions. Dysregulated circRNA expression has been linked to diseases including different types of cancer. Cancer progression is known to result from the dysregulation of several molecular mechanisms responsible for the maintenance of cellular and tissue homeostasis. The dysregulation of these processes is defined as cancer hallmarks, and the molecular pathways implicated in them are regarded as the targets of therapeutic interference. In this review, we summarize the literature on the investigation of circRNAs implicated in cancer hallmark molecular signaling. First, we present general information on the properties of circRNAs, such as their biogenesis and degradation mechanisms, as well as their basic molecular functions. Subsequently, we summarize the roles of circRNAs in the framework of each cancer hallmark and finally discuss the potential as therapeutic targets.

## 1. Circular RNAs (CircRNAs): Discovery, Biogenesis, and Degradation

### 1.1. Circular RNA Discovery

Originally, circular RNAs (circRNAs) were identified in eukaryotic cells in 1979 after the same group’s previous discovery that some viral RNAs exist in circular form by using electron microscopy [1]. CircRNAs were initially considered as transcription noise devoid of any functionality. The advent of next-generation sequencing (NGS) in the early 2000s revealed the importance of noncoding transcriptome. CircRNAs were identified as noncoding RNA species with increased stability due to their circular nature, which fostered considerable investigation of their roles in the biological processes and disease [2,3].

### 1.2. CircRNA Biogenesis and Properties

CircRNAs are now known to be transcribed from protein-coding genes and further processed by unconventional pre-mRNA splicing mechanism, referred to as backsplicing, in which the 3′-end of an exon is ligated to the 5′-end donor splice site of the same or an upstream exon [3,4]. The conventional spliceosome and canonical splicing sites are essential for the process of backsplicing. Indeed, the process of backsplicing competes with conventional linear splicing [5]. Based on their structure derived from different modes of biogenesis, circRNAs can be classified as exonic (ecircRNAs), exon-intronic (EIciRNAs), circularized intronic (ciRNAs), and tRNA intronic (tricRNAs), with the former constituting 85% of all circRNAs. In the process of circRNA biogenesis, the distant donor and acceptor splice sites are selectively brought into proximity to be ligated, and three major models accounting for it have been delineated:In a lariat-driven circularization, the exon-skipping linear splicing event produces a lariat that includes the skipped exons. Such lariat structure can further undergo internal splicing generating EIciRNAs or ecircRNAs (Figure 1A).In intron pairing-driven circularization, also known as direct backsplicing, the splice sites are brought to close proximity by complementary pairing between inverted repeats (e.g., Alu repeats) flanking circularized exons (Figure 1B).In RNA-binding protein (RBP)-driven circularization, the splice sites in intronic sequences are bridged together by certain trans-factor RBPs (Figure 1C).

Given the lower efficiency of backsplicing as compared to linear splicing, the production rate of circRNAs is lower than that of their linear counterparts. However, the lack of free ends renders them resistant to the exonuclease-mediated degradation which results in longer half-life and higher number within cells [6]. Based on this property, circRNAs can be used to serve as novel diagnostics markers or disease targets. In 2019, it was first reported that circRNAs undergo biodegradation by RNase L global RNA-degrading endonuclease [7]. In such a mechanism, circRNAs were found to be stabilized by binding to inactive unphosphorylated protein kinase R (PKR) that blocked circRNAs’ accessibility to RNase L [7,8]. Phosphorylation of PKR activates it and mediates its dissociation from circRNAs, which, in turn, exposes them to active RNase L [7] (Figure 2). Similarly, another study showed that the highly structured nature of circRNAs ensures their degradation via binding to RNA-binding protein UPF1 and its associated protein G3BP1, which causes subsequent degradation by RNase P or RNase MRP [9]. Therefore, we can conclude that the cell achieves circRNA homeostasis by possessing a well-controlled circRNA biogenesis and degradation systems (Figure 2). Dysregulation of these processes can lead to cellular aberrations and disease. Numerous transcriptome analysis studies identified more than 20,000 circRNAs expressed from 60% of total genes in eukaryote cells. Intriguingly, a portion of circRNAs is expressed in a tissue- or developmental stage-specific manner indicative of temporal and spatial control of their expression. CircRNAs have been identified in numerous species by different high-throughput RNA deep sequencing projects. In one of the studies, it was identified that around 14.4% of human fibroblast circRNAs can be linked to 69 murine testis orthologous circRNAs [10]. Such orthologous conservation features of circRNA imply the importance of their role in gene regulation.

## 2. Molecular Functions of CircRNAs

CircRNAs have been proven to be involved in various biological processes and mechanistically have been widely characterized to conduct their functions by several major mechanisms:One of the most investigated functions of circRNAs is their ability to indirectly control gene expression through sponging of microRNAs (miRNAs) [6] (Figure 1D). CircRNAs often contain miRNA response elements (MREs) which can bind miRNAs, thus preventing them from silencing their gene targets [11,12]. By such mechanism, these circRNAs are classified as competing endogenous RNAs (ceRNAs) [11]. Among the most prominent of them is ciRS-7 (CDR1as), which was discovered to possess up to 70 target sites for miR-7 [6]. Therefore, a vast number of studies has been focused on investigating ceRNA mode of action of circRNAs. However, large-scale screening studies indicate that usually circRNAs contain a limited number of MREs, indicative that this is not the only bona fide mechanism.CircRNAs can associate with RNA binding proteins (RBPs) and thus mediate different aspects of their functionality (Figure 1E). For instance, circRNAs can act as scaffolds facilitating the formation of protein complexes. For example, circ-FOXO3 was shown to interact with both MDM2 and p53, which promoted MDM2-dependent ubiquitination and degradation of p53 [13]. The same circRNA was also shown to form ternary complex with CDK2 and p21, preventing the association of the former into functional complexes with cyclins A and E, thus acting by decoy mechanism [14].CircRNAs that retain the intronic sequences of their parental genes, such as ciRNAs and EIciRNAs, are predominantly localized in the nucleus. As was shown by Pol II CLIP, EIciRNAs such as circEIF3J and circPAIP2 could associate with RNA polymerase II and enhance their parental gene expression in cis in a U1 snRNP-dependent manner [15] (Figure 1F). The broad-scale effects of circRNAs on pre-mRNA splicing are not widely characterized as yet; however, it has been known that circularization and linear splicing compete with each other on a co-transcriptional level (Figure 1G). Such regulatory mechanism was demonstrated for MBL/MBNL1 gene encoding muscleblind (MBL) splicing factor. The protein product of this gene was shown to directly promote circularization of the exon 2 of its host gene by direct binding to the flanking sequences in the adjacent introns to produce circMBL, concomitantly reducing linear splicing [5]. Finally, in addition to regulating mRNAs at transcriptional and splicing levels, circRNAs were also shown to regulate the stability of mRNAs. For example, the degraded fragments of CDR1, as circRNA could downregulate CDR1 by promoting degradation via antisense pairing [16]. On the contrary, circRasGEF1B could enhance the stability of its target ICAM1 mRNA [17].Although circRNAs have traditionally been regarded as noncoding RNAs given the lack of the 5′ cap that is normally required for the initiation of translation, the discovery of internal ribosome entry sites (IRES) within multiple circRNAs suggests that they may be extensively translated into peptides or proteins in 5′ cap-independent manner [18] (Figure 1H). One prominent example of such circRNAs is circ-ZNF609, whose IRES-dependent translation was shown to play an important role in myogenesis [19]. In addition, it was shown that initiation of translation in circRNAs can be driven by m6A methylation. CircRNAs were found to be enriched in consensus m6A motifs, whose methylation by METTL3/14 methyl transferase complex promoted the translation, and demethylation by FTO demethylase inhibited the translation [20]. In such a mechanism, m6A deposited on circRNAs could be recognized by YTHDF3 m6A reader protein, which in turn recruited eIF4G2 initiation factor [20].

## 3. Molecular Homeostasis Signaling in Cancer Hallmarks

In normal conditions, molecular homeostasis culminates at eradicating the biological destabilization brought about by concerted genetic and nongenetic physiological dysregulations. In contrast, disease develops when molecular homeostasis cannot be achieved, irrespective of however minute the physiological dysregulations are. Excessive expression or degradation of a particular circRNA can potentially affect its biological functions and overall wellbeing of an organism. Pathologically, the overexpression or downregulation of several circRNAs have been linked to the biogenesis of various diseases such as cardiovascular diseases [21], neurological diseases [21,22], osteogenic malformation [22], and several types of cancers [23,24,25].

During cancer development, scientists have identified a group of cellular qualities which progress simultaneously in order to convert a normal cell into malignant state. These qualities were defined as cancer hallmarks, and initially included such features as proliferation independent of growth signals, insensitivity to anti-growth signals, evasion of programmed cell death, limitless replication potential, promotion of angiogenesis, tissue invasion, and metastasis [26,27]. Later, this list of basic cancer hallmarks was updated with other important oncogenic features such as deregulated metabolism, ability to evade the immune system, genome instability, and tumor-promoting inflammation.

At its very least, various intracellular molecular pathways were found to be connected to ultimate cancer hallmark events [27]. Meanwhile, in studying cancer hallmarks and their associated molecular mechanisms, every biomolecule ranging from proteins, DNA and all types of RNA have been studied. Likewise, coding and noncoding RNAs have been implicated in tumorigenesis due to their ability to interfere with gene products or signaling in neoplastic cells [28,29]. In the current review, we specifically focus on circRNAs and their implication in modulating molecular signaling supporting cancer hallmarks. Considering the nature of their relationship with cancer pathogenesis, we can broadly classify circRNAs into two main groups, namely oncogenic and tumor suppressor circRNAs. In normal homeostasis conditions, tumor suppressor circRNAs are highly expressed, while the expression of oncogenic circRNAs is maintained at their lowest levels. In contrast, for cancer to grow, oncogenic circRNAs are overexpressed, while tumor suppressor circRNAs are downregulated [23,30]. Such events are governed by the disturbance of balance between circRNA biogenesis and degradation mechanisms (Figure 2).

Nowadays, a genome-wide approach is widely employed to identify cancer-specific circRNAs. In a recent study in the human tissues, 895 circRNAs have been identified as tissue-specific, in contrast to 2329 less tissue-specific and 1469 ubiquitously expressed circRNAs [31,32]. On the basis of such systematic analyses, cancer-related circRNAs can also be easily identified from the MiOncoCirc database (https://mioncocirc.github.io/) [32] (Figure 3). By comparing the expression profile between cancer and the respective noncancerous tissues, the primary functions of circRNAs can be simply classified into oncogenic and tumor suppressive modes [32].

Since cancer pathogenesis is often considered in the framework of cancer hallmark-associated pathways [26,27,33], the subsequent sections will be dedicated to the summary of circRNAs implicated in each hallmark.

### 3.1. CircRNAs and Sustaining Proliferative Signaling

CircRNAs have been implicated in various molecular pathways that promote uncontrolled cell proliferation. CiRS-7 (also known as CDR1as) is one of the most studied circRNAs that has been shown to act as an oncogenic factor in various cancers through the mechanism of miRNA sponging [34,35,36]. CiRS-7 harbors more than 70 binding sites for miR-7, a known tumor suppressor, and its expression level has been associated with the progression of such malignancies as colorectal cancer [37], lung cancer [38], and hepatocellular carcinoma [39], among others. Indeed, the implication of ciRS-7 in various types of cancer has been extensively reviewed [40]. Mechanistically, the consequence of sponging of miR-7 results in sustaining proliferative signaling by such mechanisms as regulating EGFR signaling [38,41], cell cycle machinery [38], NF-κB pathway [42], PI3K/Akt signaling [43].

Among other circRNAs regulating proliferation, circ-AMOTL1 was shown to exhibit tumorigenicity by interacting with c-Myc, thus increasing its nuclear retention and promoting stability [44]. In pancreatic ductal adenocarcinoma (PDAC), circ-PDE8A was shown to regulate MET oncogene via miR-338, thus stimulating invasive growth. Interestingly, circ-PDE8A was shown to be secreted through exosomes, and its expression correlated with the progression in PDAC patients [45].

The downregulation of various tumor-suppressor circRNAs has been associated with the increased cancer proliferation. For example, the suppression of circITGA7 was shown to promote colorectal cancer growth and metastasis [46]. Mechanistically, circITGA7 upregulates the expression of its parental gene, ITGA7, which is a negative regulator of proliferative Ras signaling. The expression of circ-FOXO3 was reported to be increased in noncancerous cells and it was downregulated in cancerous tissues. The suppression of circ-FOXO3 promotes cancer proliferation as circ-FOXO3 binds with CDK2 and p21, thus inhibiting the cell cycle progression [14]. CircZKSCAN1 (hsa_circ_0001727) and the parental mRNA, ZKSCAN1, were shown to be downregulated in hepatocellular carcinoma, and the knockdown of both of these forms promoted proliferation [47]. In addition, circZKSCAN1 was shown to modulate tumor-suppressive properties by sponging miR-873-5p, miRNA that targets DLC1, another tumor suppressor in hepatocellular carcinoma [48].

### 3.2. CircRNAs and Sustained Angiogenesis

The tumor growth is often associated with new blood vessel formation to facilitate the growing metabolic demands of the cancer cells [49]. The release of pro-angiogenic factors by the cancer cells, including VEGF, FGF, angiopoietins, and TGF-beta, has been associated with enhanced angiogenesis in various cancer cells [50]. Several circRNAs have been reported to be implicated in oncogenic angiogenesis.

CircASH2L was shown to be involved in tumorigenesis while promoting angiogenesis in ovarian cancer through miR-665-mediated regulation of VEGFA [51]. Similarly, circRNA-MYLK was shown to activate VEGFA/VEGFR2 signaling pathway through the regulation of miR-29a in bladder cancer [52]. In colorectal cancer, circ-001971 was shown to relieve miR-29c-3p-mediated inhibition of VEGFA [53]. Based on the exosomal circRNA profiling, circ-CCAC1 was demonstrated to be upregulated in cancerous extracellular vesicles and tissues of cholangiocarcinoma. Mechanistically, circ-CCAC1 acted by sponging miR-514a-5p, thus inducing the upregulation of Yin Yang 1 (YY1) transcription factor. Due to the high extracellular availability of circ-CCAC1, the uptake of circ-CCAC1 by endothelial monolayer cells disrupted endothelial barrier and induced angiogenesis [54]. Similarly to circ-CCAC1, the high exosomal content of circRNA_100338 in hepatocellular carcinoma was shown to be involved in angiogenesis and vasculogenic mimicry as was demonstrated by human umbilical vein endothelial cell (HUVEC) model [55].

### 3.3. CircRNAs and Enabling Replicative Immortality

Normally, somatic cells can undergo a limited number of cell divisions due to telomere shortening. However, cancer cells possess telomerase activity, which allows them to divide indefinitely. This property is shared with tissue-specific stem cells, which normally replenish differentiated cell populations within tissues. Recently, it became increasingly clear that tumors contain a subpopulation of stem cells, referred to as cancer stem cells (CSCs), and their abundance is linked to tumor malignancy.

The broad-spectrum implications of circRNA dysregulation in stemness-related properties in cancer are still being investigated. The key regulators of stemness, such as Oct4, SOX2, KLF4, and NANOG, have been linked to the existence of CSC population. The high expression levels of these regulators were associated with tumor metastasis and therapeutic resistance [56,57]. CircBIRC6 has been shown to regulate stemness properties by participating in the pluripotency circuitry in human embryonic stem cells (hESCs) [58]. In cancers, the expression of various circRNAs has been reported to affect the expression of the key regulators of stemness. The overexpression of circEPHB4 (hsa_circ_0081519) was shown to induce the upregulation of Oct4 and NANOG in glioma [59]. CircVRK1 was shown to negatively regulate Oct4, Sox2, and NANOG expression in breast carcinoma CSC population [60]. Similarly, the overexpression of circ_POLA2 (hsa_circ_0022812) was reported to increase the expression levels of Oct4 and NANOG in lung cancer cells [61]. However, it is worth noting that these studies did not explore the direct regulation of the reported stemness markers by circRNAs.

Various reports have highlighted the effect of circRNAs on other regulators of stemness through ceRNA mechanism. CircPTN (hsa_circ_0003949) was shown to sponge miR-145-5p, thus regulating the expression of SOX2 in glioma stem cells [62]. By sponging miR-4262 and miR-185-5p, circAGFG1 was shown to regulate stemness properties in colorectal cancer via the regulation of CTNNB1 resulting in activation of Wnt/β-catenin pathway [63]. Hsa_circ_001680 was shown to affect the expression of BMI1 via miR-340, resulting in the induction of stemness characteristics in colorectal cancer cells [64]. Our previous work has also shown that two circRNAs, hsa_circ_0082096 and hsa_circ_0066631, collectively sponge various miRNAs that are involved in multiple pathways regulating pluripotency in colorectal cancer stem cell population [65].

Besides acting as miRNA sponges, some circRNAs have also been shown to regulate stemness by other mechanisms. CircGprc5a (hsa_circ_02838) was shown to regulate stemness-related properties by producing an 11-amino acid peptide, which can bind to GPCR5A protein, facilitating its regulation of stemness [66]. Similarly, circLgr4 was shown to produce a peptide that regulates Wnt signaling pathways in colorectal cancer stem cell population [67]. CircZKSCAN1 binds with FMRP protein which results in the disruption the transcriptional activities of Wnt signaling pathway in hepatocellular carcinoma CSCs [68].

### 3.4. Resisting Cell Death: CircRNA Roles in Cancer Autophagy and Programmed Cell Death

#### 3.4.1. CircRNAs and Autophagy

Autophagy literally means “self-eating” and is an evolutionarily conserved process in eukaryotes. Autophagy pathway and ubiquitin-proteasome system (UPS) govern the balance between protein synthesis and degradation [69]. More emerging results indicate that several circRNAs may be involved in autophagy regulation through various pathways. For example, circ-DNMT1 enhances breast tumor growth by interaction with p53 and AUF1, which promotes their nuclear translocation. Circ-DNMT1-mediated nuclear translocation of p53 and AUF1 reduces the mRNA stability of DNMT1 and reciprocally reduces the expression of p53 [70]. Similarly, circCDYL also enhanced breast cancer cell proliferation by autophagy enhancement through serving as a miRNA sponge. The interaction between miR-1275 and circCDYL abrogated the inhibitory effect on ATG7 and ULK1 gene repression [71]. On the other hand, the inhibition of circHIPK3 in lung cancer-enhanced autophagy by sponging miR-124-3p resulted in the activation of STAT3 and further phosphorylation of PRKAA in STK11-dependent manner [72]. CircRAB11FIP1 was shown to promote autophagy in ovarian cancer by targeting miR-129 as well as directly interacting with DSC1 protein. In addition, circRAB11FIP1 could directly interact with FTO mRNA to enhance its translation to promote ATG5/7 mRNA methylation [73].

#### 3.4.2. CircRNAs and Programmed Cell Death

CircRNAs can contribute to tumorigenesis by inhibiting different types of programmed cell death, including apoptosis, necroptosis, ferroptosis, and pyroptosis. Circ-MAPK4, initially identified from clinical glioma samples, represses apoptosis through the reduction of PARP11 and caspase 3/7/9 cleavage [74]. Intriguingly, circ-MAPK4 maintains the p38/MAPK survival pathway signaling to sustain cell viability by hybridizing with miR-125a-3p. Such circ-MAPK4-miR-125a3p-p38/MAPK axis is essential for glioma tumorigenesis [74]. miR-382-5p-targeting circRNA, circRNA-UBAP2, promotes tumor growth by the enhancement of cell proliferation and apoptosis in ovarian cancer [75]. CircRNA-UBAP2 reverses the original role of miR-382-5p in the regulation of PRPF8 expression to upregulate cell proliferation-related genes, anti-apoptotic Bcl-2, and downregulate pro-apoptotic Bax and caspase-3 [75]. Ferroptosis has recently been identified as a distinct type of regulated cell death characterized by the accumulation of abnormal oxidized lipids and ROS in an iron-dependent manner. CircRNA cIARS was shown to physically interact with ALKBH5 RBP in hepatocellular carcinoma, which resulted in the stabilization of BCL-2/BECN1 complex and subsequent enhancement of ferroptosis [76]. On the contrary, circIL4R was shown to act as an miR-541-3p sponge to inhibit ferroptosis by the enhancement of GPX4 expression [77].

### 3.5. CircRNAs and the Activation of Invasion-Metastasis Cascade

Metastasis is another prominent cancer hallmark defined by the migration of cancer cells from the primary tumor site to different tissues and organs, which results in the development of secondary and tertiary tumor foci [78]. Typically, the cascades of epithelial-mesenchymal transition (EMT) instigate cancer cell plasticity via acquisition of mesenchymal-like phenotype that promotes invasiveness, drug resistance properties, and enhanced migration ability [79].

Owing to the fact that circRNAs can sequester miRNAs to regulate gene expression, many studies have corroborated the role of circRNA–miRNA–mRNA networks in promoting or inhibiting invasion-metastasis cascade via EMT [80]. In a recent study, circBFAR was discovered to promote proliferation and migration in pancreatic ductal adenocarcinoma. Mechanistically, circBFAR was shown to repress the function of miR-34b-5p which normally inhibits the MET gene expression. In the same study, it was shown that the suppression of circBFAR-mediated tumorigenicity and metastasis properties in in vivo model [81]. In another study, circ_0001666 was found to facilitate EMT, cell proliferation, and invasion in pancreatic cancer. Mechanistically, the authors delineated a novel circ_0001666/miR-1251/SOX4 regulatory axis in pancreatic cancer cells, whereby the oncogenic effects of circ_0001666 were rescued by miR-1251 overexpression and inhibited SOX4 expression in both in vitro and in vivo experiments [82]. Another recent study demonstrated the role of circST6GALNAC6 in inhibiting EMT/metastasis in breast cancer by sponging miR-200a-3p, which promoted STMN1 gene expression. STMN1 was shown to inhibit EMT and metastasis in cancer progression. Furthermore, the overexpression of circST6GALNAC6 significantly inhibited EMT and invasion properties in vitro and also attenuated breast cancer growth in vivo [83].

Apart from miRNAs, circRNAs also regulate target mRNAs and their translation by sequestering RNA binding proteins (RBPs) [84]. However, little is known on the role of circRNA-RBP interactions in promoting invasion and metastasis in cancer [85]. In a recent study, a novel circRNA, circ0005276, was shown to enhance cell proliferation, EMT, and migration in prostate cancer by interacting with FUS RBP. The binding of circ0005276 with FUS activates X-linked inhibitor of apoptosis protein (XIAP), which is also the host gene of circ0005276. In fact, circ0005276-FUS positively regulated XIAP transcription which drove metastasis in in vivo animal model [86]. In lung adenocarcinoma (LUAD), circXPO1 binding with IGF2BP1 enhanced the stability of CTNNB1 mRNA that encodes β-catenin crucial for EMT. This contributed to tumor progression with 51% of resected LUAD tumor samples expressing β-catenin. The silencing of circXPO1 in patient-derived xenograft (PDX) model inhibited LUAD growth by reducing its metastatic properties [87].

To sum up, an increasing number of studies have elucidated the importance of circRNAs and their association with miRNAs and RBPs in mediating invasion-metastasis cascade, resulting in promoting or inhibiting tumor progression in both in vitro and in vivo models.

### 3.6. CircRNAs and Insensitivity to Anti-Growth Signals

During cancer progression, tumor suppressor genes (TSGs) are often inactivated and fail to halt indefinite cell proliferation, induce senescence, apoptosis, and eventually inhibit abnormal cell transformation [88]. Several crucial TSGs have been described as essential checkpoint regulators preventing cell proliferation in abnormal conditions favoring oncogenesis; among them are RB1, TP53, PTEN, CDH1 [89], and APC [90,91,92]. These TSGs play an important role in genome stabilization by activating apoptosis, suppression of cell cycle activity, and induction of DNA repair mechanisms. Besides the mutations inactivating these TSGs, their functions can also be interrupted when aberrantly expressed oncogenic proteins bind to these suppressors, causing their degradation or inactivation [88].

On this note, various circRNAs have been linked to either inactivating TSGs or post-transcriptionally directly or indirectly repressing TSGs expression [93,94]. Circ-ITCH downregulation was identified in bladder cancer (BCa) and discovered to sponge miR-17/miR-224 which regulate p21 and PTEN expression [95]. Another study also reported that circ-ITCH ectopic overexpression was sufficient to reduce triple negative breast cancer (TNBC) growth in vitro and in vivo [96]. Furthermore, BCa proliferation was inhibited by circRNA-3 (BCRC-3) through miR-182-5p/p27 signaling pathway [97]. Likewise, in gastric cancer, cell cycle progression was discovered to be inhibited by circ-YAP1 by targeting p27 Cdk inhibitor [98]. Suppression of apoptosis is another salient feature of oncogenic transformation that can occur due to evading such crucial TSGs as p53 [99]. For example, Lou et al. reported that CDR1as downregulation in glioblastoma supports p53 inhibition and hence glioblastoma progression increase [100]. The authors showed that inhibition of p53/MDM2 complex by CDR1as was able to suppress glioblastoma growth [100]. CircRNAs interference with apoptotic pathway did not only promote cancer cell survival and proliferation; studies have also reported cancer drug resistance mediated by apoptotic pathway suppressing circRNAs [101,102]. For instance, circAKT3 induced cisplatin resistance by suppressing apoptosis effector caspase 3 cleavage [102]. Furthermore, 5-fluorouracil (5-FU) chemoresistance was reported in breast cancer cell upregulating CDR1as through caspase-dependent apoptosis inhibition [101]

### 3.7. CircRNAs and Evading the Immune System

Tumor cells are capable of escaping and debilitating the immune system recognition and attack through the formation of a specialized immunosuppressive microenvironment or other immune escape mechanisms. For instance, hsa_circ_0020397 was reported to upregulate TERT and PD-L1 activity via sponging of miR-138 in colorectal cancer cells. Whereas TERT is a telomerase that immortalizes cancer cells by stabilizing telomere length [103], PD-L1 is an immune checkpoint protein that interacts with PD-1 to induce cancer immune escape [104,105]. Alternatively, upregulated PD-L1 is capable of directly inhibiting T cell activity and apoptosis of cancer cells without binding to PD-1 [106]. CircFGFR1 was shown to interact with miR-381-3p, thus elevating the expression of its target gene, *CXCR4*, consequently promoting NSCLC progression and resistance to anti-PD-1-based therapy [107]. Circ-CPA4 targeting of let-7 miRNA/PD-L1 axis was shown to regulate NSCLC progression as well as the biological functions of tumor-associated immune cells [108]. CircAMOTL1 has been shown to act as an RBP sponge by interacting with STAT3 and DNMT3A, which are both the targeted by miR-17-5p. Normally, DNMT3A downregulates miR-17-5p expression by methylating the promoter of *MIR17* gene, leading to the reduction of miR-17-5p level, which in turn results in the upregulation of STAT3, which modulates tumor immunity regulation [109,110]. In addition, novel aberrant fusion circRNAs (f-circRNAs) generated as a result of cancer-associated chromosomal translocations may contribute to immunocyte-mediated tumor immune response [111].

### 3.8. CircRNAs and Deregulated Cellular Energetics

The metabolism of cancer cells was found to be largely dependent on glycolysis rather than the oxidative phosphorylation (OXPHOS) respiration system, thus allowing them to generate energy irrespective of oxygen availability [112]. The Warburg effect allows cancer cells to use glycolytic products to drive their anabolic metabolism including biosynthesis of nucleic acid, lipids, and amino acids [113]. During cellular stress, AMP-activated protein kinase (AMPK) senses cellular energy status to regulate glycolysis, lipolysis, and other cellular metabolism by shutting down anabolism and initiating catabolism [114]. On this note, Li et al. reported that circACC1 regulates the assembly and activation of AMPK in colorectal cancer [115]. Similarly, circRNAs have also been found to participate in modulating the energy metabolism in cancer cells. For instance, Zhang et al. recently discovered that hsa_circ_AKAP7 and other four circRNAs enriched in prostate cancer interfered with lipid metabolism pathway [116]. Likewise, circMAT2B was also found to regulate glycolysis in hepatocellular carcinoma [117]. In conclusion, the aforementioned studies indicate that circRNAs are potent participants in cellular energy deregulation promoting this hallmark’s development.

### 3.9. CircRNAs and Genomic Instability

Genomic instability and its associated mutagenesis have been linked to the development of cancer. A few studies highlighted the potential role of circRNAs in regulating the genomic stability. CircSMARCA5 was found to form an R-loop with its parental gene, SMARCA5, resulting in transcriptional pausing and the production of truncated mutant protein [118]. M6A RNA methylation has been proven to regulate various biological pathways. For instance, m6A modification of circNSUN2 promotes its transport to the cytoplasm, where it forms a complex with IGF2BP2 protein. The circNSUN2/IGF2BP2 complex stabilizes HMGA mRNA, which results in the enhancement of metastasis in colorectal carcinoma [119]. FECR1 circRNA was shown to recruit TET1 demethylase to the promoter of FLI1 gene and downregulate DNMT1 methyltransferase to reduce DNA methylation. This resulted in the upregulation of FLI1 gene expression leading to metastasis in breast cancer [120]. Apart from the mechanisms of DNA or RNA methylation, another circRNA, called circSMARCA5, was shown to form DNA:RNA hybrid to terminate the parental SMARCA5 gene transcription early. The lower SMARCA5 expression induced by circSMARCA5 attenuated DNA repair capacity to enhance drug sensitivity [121]. Additionally, circAKT3 was shown to be involved in maintaining the genomic integrity by promoting DNA repair mechanism in gastric cancer [102]. Interestingly, some circRNAs are produced from translocated or fusion genes. The translocation of PML/RAR, a common translocation in acute promyelocytic leukemia patients, produced a fusion circRNA that contributed to various cancerous hallmarks, including progression of cancer and resistance to therapy [122]. Similarly, in lung cancer, an oncogenic fusion of EML4/ALK generated a fusion circRNA, F-circEA-2a, which was shown to exhibit tumor promotion properties in lung cancer development [123].

### 3.10. CircRNAs and Inflammation

The involvement of tumor microenvironment (TME) in the development and progression of cancer has been well established [124,125]. Tumor-infiltrating innate and adaptive immune cells have been discovered to create an inflammatory TME which supports cancer growth [126]. CircRNAs have been found to regulate immune cells in order to initiate inflammatory TME in tumors [127]. For example, Shi et al. reported circARSP91 suppression in hepatocellular carcinoma (HCC), circRNA that is known to enhance natural killer (NK) cells immune surveillance [128]. The expression of inflammatory cytokines IL-13 and IL-6 was increases by hsa_circ_0005519 targeting hsa-let-7a-5p in CD4+ T cells [129]. In addition, circRNA hsa_circ_0012919 supports epigenetic regulation of CD70, a member of the TNF family, which promotes the activation, proliferation, and differentiation of T cells and B cells [127,130]. Wang et al. also discovered that hsa_circ_0064428 might be the regulator of tumor-infiltrating leukocytes (TILs) and affect HCC prognosis [131]. In summary, circRNAs affect the combination of inflammatory cytokines activity, and TME immune cells cooperate to create the inflammatory conditions that support tumor progression.

## 4. CircRNA as Therapeutics Molecular Pathway Targets in Cancer

RNA therapeutic technology provides the opportunity of targeting every coding and noncoding gene in biological systems. Interestingly, RNA therapeutics efficacy has been proven to be workable by few approved RNA drugs for liver, muscle, and rare genetic diseases [132]. Considering that many circRNAs were confirmed to promote cancer progression, targeting them offers a great potential therapeutic strategy for combating various cancers [27].

Overall, circRNA studies currently focus more on circRNA differential expression in cancers based on the premises that circRNAs can serve as more reliable and accessible liquid biomarkers for cancer detection [133]. Other than the diagnostic value, targeting circRNA-mediated molecular pathways for therapeutic purposes is also conceivable if the full molecular mechanisms of action of the most dysregulated circRNAs implicated in cancers are known. Currently, this perspective still remains unrealized in clinical practice [84]. Therefore, gain-of-function and loss-of-function strategies are deployed in circRNA research in order to harness dysregulated circRNAs for cancer therapy [84]. Precisely, tumor suppressor circRNAs are ectopically expressed in order to replenish the downregulated tumor suppressor circRNAs in cancers [134]. For example, several transient circRNA overexpression DNA vectors harboring the Alu repeats that facilitate circularization have been developed [4,43,135]. Likewise, lentivirus and AAV delivery systems have also been used to generate stable circRNA ectopic expression in vitro and in vivo [136].

In contrast, oncogenic circRNAs are normally targeted for degradation using several techniques. For instance, RNA interference (RNAi) methods represent the most prominent circRNA degradation strategy currently available. Researchers have used both the conventional short interference RNA (siRNA) [24,137] and the more lasting short hairpin RNA (shRNA) techniques to degrade copious oncogenic circRNAs [138,139].

In addition to siRNAs and shRNAs, clustered regularly interspaced short palindromic repeats (CRISPR)/CRISPR-associated protein (Cas) systems represent an effective and promising approach for targeting circRNAs. Whereas CRISPR/Cas9 is normally used to edit DNA [140], CRISPR/Cas13 is a recently discovered system that has been utilized to target RNA [141]. At present, both systems have been used to modulate the expressions of circRNAs [134,142]. In one of the studies, CRISPR/Cas9 was used to delete the complementary Alu repeat in HIPK3 gene, thus preventing the expression of circHIPK3 [72]. Similarly, circRNA CDR1as was also downregulated using CRISPR/Cas9 and gRNA targeting CDR1as locus [35]. This evidence proved the possibility of therapeutic targeting of circRNAs using the CRISPR/Cas9 system; however, the concerns are raised about affecting the linear counterparts of circRNAs which can lead to off-target effects [84]. Therefore, various CRISPR/Cas13 orthologs have been investigated for circRNA targeting. In a recent study, CRISPR/Cas13d was used to downregulate oncogenic circFAM120A [84]. Similarly, circZKSCAN1 was also effectively downregulated using CRISPR/Cas13d system [143]. More detailed information on CRISPR/Cas system-mediated targeting of circRNAs can be found in recent reviews [84,134].

## 5. Conclusions and Perspectives

The current review summarizes the functions of various circRNAs in different types of cancer (Figure 4). CircRNAs represent a unique class of RNA molecules characterized by increased stability due to their circular nature, which have been demonstrated to exert their biological functions by such mechanisms as miRNA sponging, modulating the activity of RBPs, regulating transcription and splicing and translation of peptides. As was originally proposed by Hanahan and Weinberg in 2000, the understanding of the process of oncogenesis could be facilitated by defining the characteristic cancer hallmarks and understanding their associated molecular pathways. To systematize the literature regarding implication of different circRNAs in cancer, we classified circRNAs according to ten currently accepted cancer hallmarks (Figure 5). These circRNAs can be regarded as potential diagnostics and therapeutic targets.

## Figures and Tables

**Figure 1 cancers-14-00862-f001:**
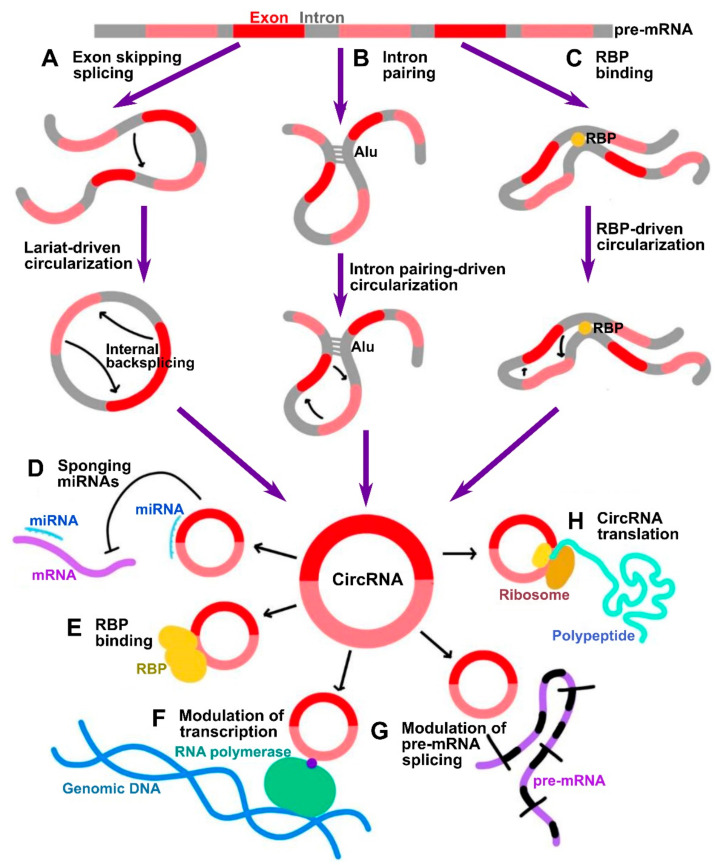
Summary of circRNA biogenesis and functional mechanisms. (**A**–**C**) Three major mechanisms of circRNA biogenesis by which distant backspliced sites are brought into proximity: (**A**) in lariat-driven circularization, circRNA is generated by internal backsplicing within a lariat formed by exon skipping direct splicing; (**B**) in intron pairing-driven circularization, backspliced sites are brought into proximity by pairing between inverted repeats such as Alu; (**C**) in RBP binding-driven circularization, backsplicing is facilitated by RBPs. (**D**–**H**) Summary of major biological functions of circRNAs: (**D**) miRNA sponging; (**E**) modulation of functions of RBPs; (**F**) modulation of transcription; (**G**) modulation of splicing; (**H**) translation of peptides.

**Figure 2 cancers-14-00862-f002:**
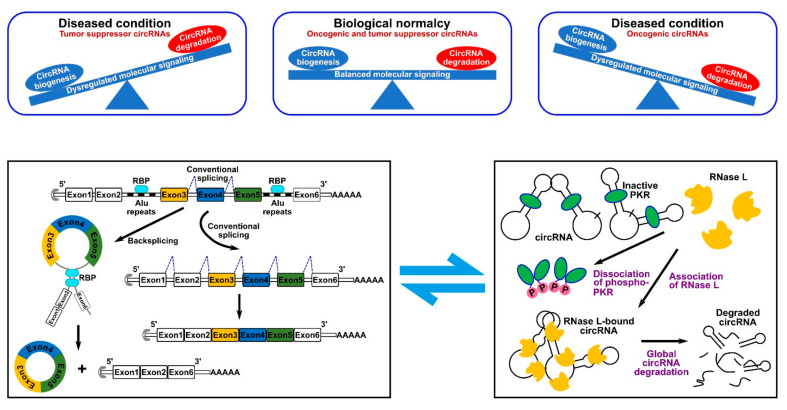
CircRNA homeostasis determines normal biological conditions or disease state. (**Top** panel) such homeostasis is determined by the fine balance between circRNA biogenesis and degradation mechanisms. (**Bottom** panel) schematic representation of a typical RBP-driven biogenesis mechanism (**left**) and of the RNase L-dependent degradation mechanism (**right**).

**Figure 3 cancers-14-00862-f003:**
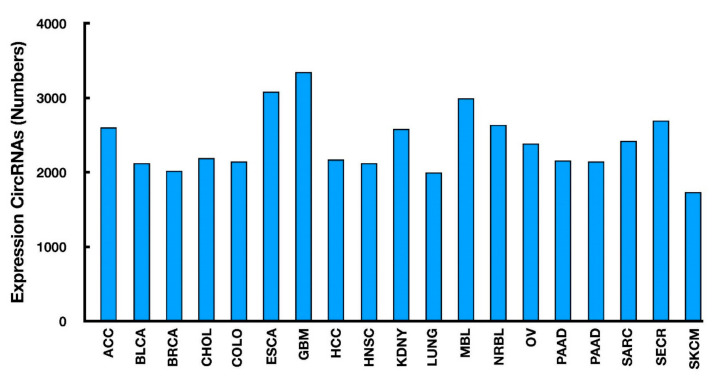
The numbers of cancer type-specific circRNAs retrieved from the MiOncoCirc database.

**Figure 4 cancers-14-00862-f004:**
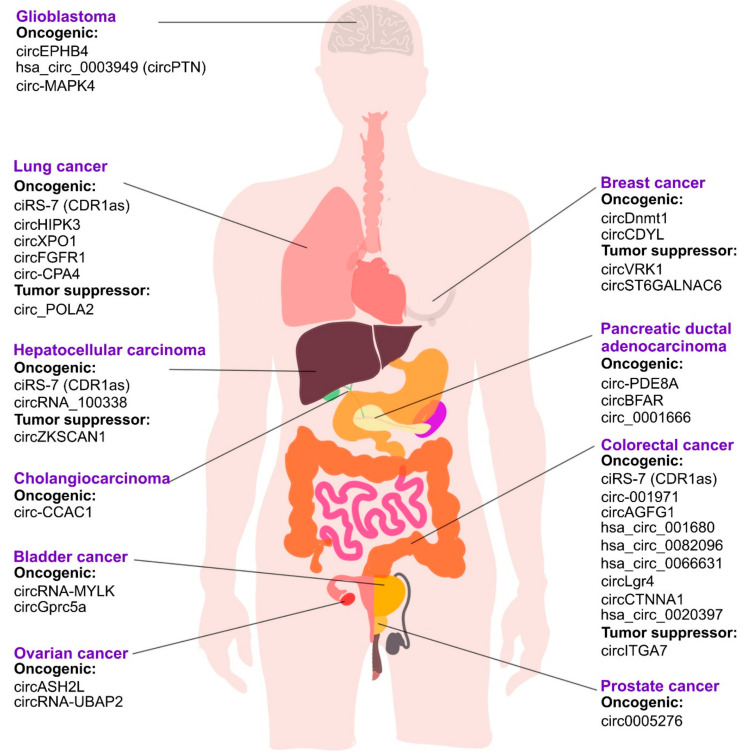
Overview of various cancer types and the selected circRNAs implicated in them with either oncogenic or tumor suppressor role.

**Figure 5 cancers-14-00862-f005:**
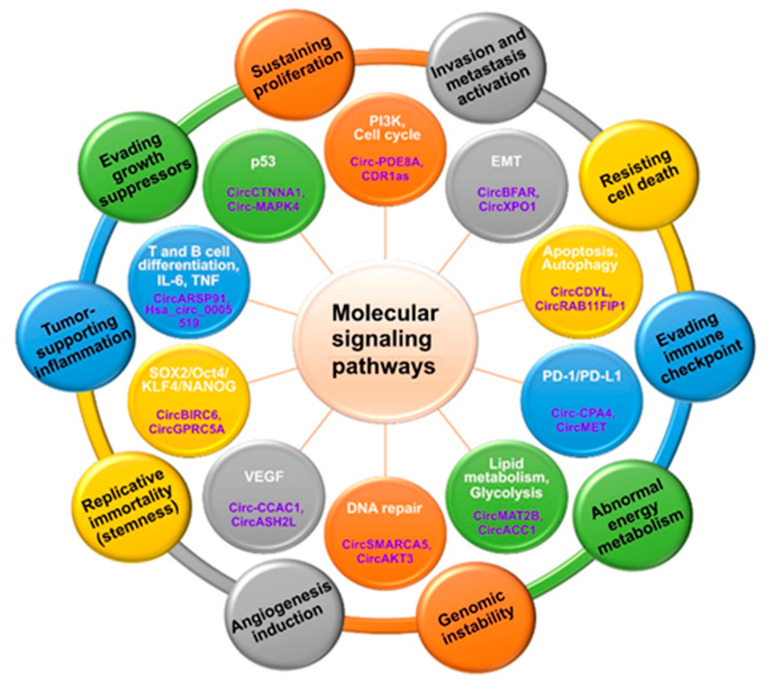
Overview of 10 cancer hallmarks (outer circles), and their major associated pathways (inner circles) with the selected implicated circRNAs.

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
