# Peer review of "Circular RNAs Modulate Cancer Hallmark and Molecular Pathways to Support Cancer Progression and Metastasis"

_cancers, 2022, doi:10.3390/cancers14040862_

Round 1
Reviewer 1 Report
This well written review summarize the literature today's available on circRNAs - non-coding sequences which nevertheless have been shown to influence biological functions of carcinogenesis. The topic is not original since it is a review, but the approach of the review summarizing the major discovered circRNA within the hallmarks of cancer is original and of help for understanding which circRNAs could be taken into account as target for new cancer treatment approaches. The structured review in dependence of the hallmarks of cancer.Authors could explain which type of search machine and which criteria they used in order to select the literature they summarize in their review.
Improve the quality of images /drawings and complete the legends of the figures
Correct punctuation mainly in legends
Author Response
Comments and Suggestions for Authors
This well written review summarize the literature today's available on circRNAs - non-coding sequences which nevertheless have been shown to influence biological functions of carcinogenesis. The topic is not original since it is a review, but the approach of the review summarizing the major discovered circRNA within the hallmarks of cancer is original and of help for understanding which circRNAs could be taken into account as target for new cancer treatment approaches. The structured review in dependence of the hallmarks of cancer.
Authors could explain which type of search machine and which criteria they used in order to select the literature they summarize in their review.
- Improve the quality of images /drawings and complete the legends of the figures
Thank you for the suggestion. We have modified the illustrative figures and improved the descriptions in the legends in the revised manuscript.
- Correct punctuation mainly in legends
We have modified the figure legends and corrected the grammar errors including the punctuation. Thanks for the reminder.

Reviewer 2 Report
This review summaries the properties of circRNAs including their biogenesis, degradation mechanisms, molecular functions, and involvements of circRNAs in the framework of each cancer hallmark. The content of this paper is substantial, trying to clarify the key role of circRNAs in the framework of cancer hallmark-associated pathways,such as the involvements of circRNAs in the sustaining proliferative signaling, invasion and metastasis activation, resisting cell death, genomic instability, etc. However, these views have been well summarized and prospected by many other scholars, so there are not many novel conclusions and views in this paper. In addition, the writing of this paper is not standardized and accurate in many places, so it needs to be carefully modified and improved.
Several suggestions for improvement:
- The declarative conclusions need to be supported by references, but the citation evidence was lacking in many places, for example, line 61,line 86 and line 491 etc.
- Line 216: “CircZKSCAN1 (hsa_circ_0001727)”, Line 260: “Hsa_circ_0003949 (circPTN) was shown………”. The description of circRNAs should be consistent.
3) In section 3.1, the authors try to prove that circRNAs were involved in the process of cancer proliferation and gives an example of CIRS-7, but the correlation between cirS-7 and cancer cell proliferation was not clearly explained in this section.
4)The meaning of each icon in the top panel in Figure1 was not fully explained in main text or legend. Does the black arrow in the figure indicate the internal splicing? The yellow solid dot means the RBP?
5)Figure 2 bottom panel: What is the meaning of the blue arrow in the picture? Is there a causal relationship between the left panel and right panel?
6)Figure3: The number text in the vertical axis should be written in the same format. For example, the “1.700” should be corrected as “1,700”.
7)The reference format was inconsistent. Most references lack journal names. It should be normalized.
Author Response
Comments and Suggestions for Authors
This review summaries the properties of circRNAs including their biogenesis, degradation mechanisms, molecular functions, and involvements of circRNAs in the framework of each cancer hallmark. The content of this paper is substantial, trying to clarify the key role of circRNAs in the framework of cancer hallmark-associated pathways,such as the involvements of circRNAs in the sustaining proliferative signaling, invasion and metastasis activation, resisting cell death, genomic instability, etc. However, these views have been well summarized and prospected by many other scholars, so there are not many novel conclusions and views in this paper. In addition, the writing of this paper is not standardized and accurate in many places, so it needs to be carefully modified and improved.
Several suggestions for improvement:
- The declarative conclusions need to be supported by references, but the citation evidence was lacking in many places, for example, line 61,line 86 and line 491 etc.
Thank you for your comments. The indicated part with their citations are listed below:
line 61
“CircRNAs are now known to be transcribed from protein-coding genes and further processed by unconventional pre-mRNA splicing mechanism referred to as backsplicing in which the 3’-end of an exon is ligated to the 5’-end donor splice site of the same or an upstream exon [1,2]”.
line 86
“Phosphorylation of PKR activates it and mediates its dissociation from circRNAs, which in its turn exposes them to active RNase L [3] (Figure 2)”.
line 491
“Precisely, tumor suppressor circRNAs are ectopically expressed in order to replenish the downregulated tumor suppressor circRNAs in cancers [4]”.
- Line 216: “CircZKSCAN1 (hsa_circ_0001727)”, Line 260: “Hsa_circ_0003949 (circPTN) was shown………”. The description of circRNAs should be consistent.
Corrected.
- In section 3.1, the authors try to prove that circRNAs were involved in the process of cancer proliferation and gives an example of CIRS-7, but the correlation between cirS-7 and cancer cell proliferation was not clearly explained in this section.
We have added specific examples of pathways/mechanisms specifically regulating sustained proliferation:
“Mechanistically, the consequence of sponging of miR-7 results in sustaining proliferative signaling by such mechanisms as regulating EGFR signaling [5], [6], cell cycle machinery [6], NF-κB pathway [7], PI3K/Akt signaling [8].”
- The meaning of each icon in the top panel in Figure1 was not fully explained in main text or legend. Does the black arrow in the figure indicate the internal splicing? The yellow solid dot means the RBP?
We have updated Figure 1 with new labels, and added more detailed description into the figure legend.
- Figure 2 bottom panel: What is the meaning of the blue arrow in the picture? Is there a causal relationship between the left panel and right panel?
Thank you for noticing. Indeed, there was a mistake in this Figure. Now we corrected it by changing to bidirectional arrow which signifies the balance between the process of biogenesis (left panel) and degradation (right panel).
- Figure3: The number text in the vertical axis should be written in the same format. For example, the “1.700” should be corrected as “1,700”.
Corrected.
- The reference format was inconsistent. Most references lack journal names. It should be normalized.
The reference format is corrected to the MDPI style.
Reference:
- Memczak, S.; Jens, M.; Elefsinioti, A.; Torti, F.; Krueger, J.; Rybak, A.; Maier, L.; Mackowiak, S.D.; Gregersen, L.H.; Munschauer, M.J.N. Circular RNAs are a large class of animal RNAs with regulatory potency. 2013, 495, 333-338.
- Kramer, M.C.; Liang, D.; Tatomer, D.C.; Gold, B.; March, Z.M.; Cherry, S.; Wilusz, J.E.J.G.; development. Combinatorial control of Drosophila circular RNA expression by intronic repeats, hnRNPs, and SR proteins. 2015, 29, 2168-2182.
- Liu, C.-X.; Li, X.; Nan, F.; Jiang, S.; Gao, X.; Guo, S.-K.; Xue, W.; Cui, Y.; Dong, K.; Ding, H.J.C. Structure and degradation of circular RNAs regulate PKR activation in innate immunity. 2019, 177, 865-880. e821.
- Holdt, L.M.; Kohlmaier, A.; Teupser, D. Circular RNAs as Therapeutic Agents and Targets. Front Physiol 2018, 9, 1262, doi:10.3389/fphys.2018.01262.
- Liu, L.; Liu, F.B.; Huang, M.; Xie, K.; Xie, Q.S.; Liu, C.H.; Shen, M.J.; Huang, Q. Circular RNA ciRS-7 promotes the proliferation and metastasis of pancreatic cancer by regulating miR-7-mediated EGFR/STAT3 signaling pathway. Hepatobiliary Pancreat Dis Int 2019, 18, 580-586, doi:10.1016/j.hbpd.2019.03.003.
- Zhang, X.; Yang, D.; Wei, Y. Overexpressed CDR1as functions as an oncogene to promote the tumor progression via miR-7 in non-small-cell lung cancer. Onco Targets Ther 2018, 11, 3979-3987, doi:10.2147/OTT.S158316.
- Su, C.; Han, Y.; Zhang, H.; Li, Y.; Yi, L.; Wang, X.; Zhou, S.; Yu, D.; Song, X.; Xiao, N.; et al. CiRS-7 targeting miR-7 modulates the progression of non-small cell lung cancer in a manner dependent on NF-kappaB signalling. J Cell Mol Med 2018, 22, 3097-3107, doi:10.1111/jcmm.13587.
- Pan, H.; Li, T.; Jiang, Y.; Pan, C.; Ding, Y.; Huang, Z.; Yu, H.; Kong, D. Overexpression of Circular RNA ciRS-7 Abrogates the Tumor Suppressive Effect of miR-7 on Gastric Cancer via PTEN/PI3K/AKT Signaling Pathway. J Cell Biochem 2018, 119, 440-446, doi:10.1002/jcb.26201.
